# Absence of Injury Is Not Absence of Pain: Prevalence of Preseason Musculoskeletal Pain and Associated Factors in Collegiate Soccer and Basketball Student Athletes

**DOI:** 10.3390/ijerph19159128

**Published:** 2022-07-26

**Authors:** Oluwatoyosi B. A. Owoeye, Jamil R. Neme, Paula Buchanan, Flavio Esposito, Anthony P. Breitbach

**Affiliations:** 1Department of Physical Therapy and Athletic Training, Doisy College of Health Sciences, Saint Louis University, St. Louis, MO 63104, USA; anthony.breitbach@health.slu.edu; 2Department of Family and Community Medicine, School of Medicine, Saint Louis University, St. Louis, MO 63104, USA; jamil.neme@health.slu.edu; 3Department of Health and Clinical Outcomes Research, School of Medicine, Saint Louis University, St. Louis, MO 63104, USA; paula.buchanan@health.slu.edu; 4Department of Computer Science, Saint Louis University, St. Louis, MO 63103, USA; flavio.esposito@slu.edu

**Keywords:** pain, recurrent injury, injury prevention, collegiate sport

## Abstract

Unlike musculoskeletal (MSK) injuries, MSK pain is rarely studied in athletes. In this study, we examined the prevalence of preseason MSK pain in apparently healthy collegiate soccer and basketball players and its relationship with previous injuries (1-year history), among other factors. Ninety-seven eligible student athletes (mean age: 20.1 (SD: 1.6) years; 43% male; 53% soccer players) completed a baseline questionnaire comprising questions related to demographics, medical and 1-year injury history and any current MSK pain and the corresponding body location. The overall prevalence of preseason MSK pain was 26% (95% CI: 17–36%) and it did not differ by sex or sport. The back (6.2%) and knee (5.2%) regions were reported to be the most frequently affected body parts for preseason MSK pain. Athletes with a previous injury and with perception of incomplete healing had 3.5-fold higher odds (OR: 3.50; 95% CI: 1.28–9.36) of baseline MSK pain compared with those without a previous injury. One in four collegiate soccer and basketball players had preseason MSK pain. Collegiate sports medicine professionals should consider conducting routine preseason evaluations of MSK pain in their athletes and initiate appropriate interventions for the prevention of MSK pain and its potential consequences among athletes.

## 1. Introduction

Pain is subjective feeling of an unpleasant sensory and emotional experience arising from actual or potential tissue damage [1]. Competitive athletes often suffer from musculoskeletal (MSK) pain that is either related to ongoing underlying tissue damage (nociceptive pain)—for example, from gradual and/or sudden onset injuries—or damage that develops after an injury (nociplastic pain)—for example, chronic pain from a traumatic injury that has not completely healed [1,2,3,4]. Pain may or may not stop athletes from their continued participation in sport; often it does not, especially if there is no physical sign of a trauma [2]. However, ongoing pain may be an indication of potentially serious underlying tissue damage and may result in a more serious future injury that impacts short- and long-term performance and health [1,5,6].

Unlike MSK injuries, MSK pain is rarely studied in athletes. Although several epidemiological studies exist describing the risk and burden of acute/traumatic injuries in team collegiate sports [7,8], evidence regarding sport-related MSK pain or chronic injuries among collegiate athletes is limited. The need for a more holistic approach to rehabilitation and return-to-play protocols after a time-loss injury in athletes inclusive of both physical and mental components has been emphasized [9]. Recent studies suggest that the rate of occurrence of gradual onset injuries is as high as that of sudden onset injuries among soccer and basketball players [3,4]. In team sports, players often accept that pain is an inherent part of the sport and are motivated by a desire to maintain their position on a roster regardless of MSK pain. Thus, several athletes have reported hiding injuries as “mere pain” to secure playing time (McCallum, 2022; PhD thesis). These motivations and urgencies could potentially lead to further pain and severe injuries that restrict movement and impact athlete performance and health [1].

A previous injury has been identified as the most consistent risk factor for new injuries in sport [5,10,11]. However, the etiology of a new injury following a previous injury is not fully understood. One potential mechanism is altered motor control following an initial traumatic sport injury. Research in this area is, however, currently limited to recurrent injury risk after an incidence of concussion [12]. There is also a likelihood that athletes with a history of injury, especially from a prior season, may have residual symptoms, including pain due to their injury not healing completely or athletes not having comprehensive rehabilitation protocols or not completing their rehabilitation protocols before returning to sports. Analyzing pain and its relationship with a previous injury may inform the individualized prevention of the reoccurrence of injuries and help protect the overall physical and mental health of athletes. In this study, we examined the prevalence of preseason MSK pain in apparently healthy collegiate soccer and basketball players and the associated factors, including reports of previous injuries with or without the perceived incomplete healing of such injuries.

## 2. Materials and Methods

### 2.1. Study Design and Participants

This study was a part of the RICHLoad (Reducing Injuries in Collegiate Athletes Through Load Management) Project. Briefly, the RICHLoad Project is an ongoing two-season prospective research study aimed at investigating the relationships between sport-related load, injuries and pain toward the development and evaluation of the RICHLoad Athlete Monitoring App. All the participants in the RICHLoad Project present at the RICHLoad baseline testing session were invited to complete a baseline questionnaire comprising questions related to demographics (e.g., age and sex), medical and injury history (time-loss) in the previous year (i.e., a previous injury) and any current MSK pain and the corresponding body location. Athletes were eligible to participate in this study if they did not have any ongoing time-loss MSK injuries and were functionally viable to complete the baseline testing.

The study outcome (the dependent variable) was MSK pain; the independent variables evaluated were previous injuries (self-report of no injury vs. yes injury with incomplete healing or yes injury with complete healing), biological sex, body weight, height and BMI. The following two questions from the baseline questionnaire were used to obtain the response categories for previous injuries, the primary independent variable of the study: “Have you had any injury requiring at least 1 day of missed participation from sport or exercise in the past 12 months?” and (if yes) “Do you feel your injury/injuries has/have healed 100%?” The study outcome and all independent variables except body weight and height (from which the BMI was calculated) were collected through the baseline questionnaire. The collection of body weight and height data was undertaken during a baseline testing session that was a part of the RICHLoad Project. This study was approved by the Saint Louis University Institutional Review Board, Saint Louis, MO, USA (#31489). Each participant gave written informed consent prior to enrollment in the study.

### 2.2. Statistical Analysis

The descriptive statistics of the mean and standard deviation, frequency and proportions with a 95% CI were used to describe the baseline characteristics and the distribution of MSK pain per body location, as relevant. A multivariable logistic regression model controlling for clusters by teams was used to examine the factors associated with MSK pain in the student athletes. All the independent variables considered in this study were entered into an initial regression model and a backward elimination method was used to remove variables if they were not statistically associated with MSK pain and if they did not confound significant associations between MSK pain and any given independent variable. The point estimates from the regression analysis were presented as an odds ratio (OR) with a 95% CI. An alpha of 0.05 was used to determine the statistical associations. Sex and sport were kept in the final model because of their clinical relevance in the MSK pain outcomes.

## 3. Results

### 3.1. Participant Characteristics

Ninety-seven student athletes (age: 20.1 (SD: 1.6) years; 43% male; 53% soccer players) completed the questionnaire related to MSK pain. Forty-four (45%; 43% male, 47% female) student athletes reported a previous injury in the previous year and fifteen (15%; 12% male, 18% female) perceived that their previous injury had not completely healed. A detailed description of the participants characteristics is presented in Table 1.

### 3.2. Preseason Musculoskeletal Pain

Twenty-five players reported at least one MSK pain, equivalent to a preseason MSK prevalence of 26% (95% CI: 17–36%); 24% (95% CI: 12–40%) in male players and 27% (95% CI: 16–41%) in female players and 29% (19% CI: 17–44%) in soccer players and 21% (95% CI: 11–36%) in basketball players. The distribution of the location of MSK pain in the student athletes is presented in Table 2. Overall, the back region (*n* = 6; 6.2%) was reported to be the most frequently affected body part for preseason MSK pain. Groin (7.8%) and quadriceps pain (7.8%) were the most common in soccer players and back pain (6.5%) was the most common MSK pain in basketball players.

### 3.3. Factors Associated with Musculoskeletal Pain

A previous injury was significantly associated with baseline MSK pain in players in the multivariable logistic regression model (Table 3). Players with a perception of an incompletely healed previous injury had 3.5-fold higher odds (OR: 3.50; 95% CI: 1.28–9.36) of baseline MSK pain compared with those without a previous injury. Although the athletes with a completely healed previous injury had lesser odds for baseline MSK pain, this was not statistically significant (OR: 0.53; 95% CI: 0.19–1.52). None of the other independent variables were associated with baseline MSK pain.

## 4. Discussion

Our primary objective was to assess the prevalence of preseason MSK pain in collegiate soccer and basketball players intending an unrestricted participation in their sport. In this study, we found that one in four athletes had baseline MSK pain prior to the commencement of a new season. Preseason MSK pain was similar for male and female players but tended to be slightly higher in soccer players when compared with basketball players. The back and knee regions were the most common locations of pain among players. Other common body locations included the groin and quadriceps. An additional objective in this study was to examine the factors associated with MSK pain in collegiate soccer and basketball athletes. Given the current evidence regarding a previous injury being a consistent predictor of a new injury [5,10,13], we wanted to know if a previous injury (classified as no previous injury), a previous injury with perceived incomplete healing (<100%) and a previous injury without perceived incomplete healing related to preseason MSK pain. Players with a perception of an incompletely healed previous injury had significantly higher odds of pain and those without a perception of an incompletely healed previous injury tended toward lower odds of preseason pain, although this was not statistically significant. It is likely that players with a perception of an incompletely healed previous injury had an altered healing process. This finding supports the explanation for the possible mechanism by which a previous injury is related to a new injury [5]. Coaches and sports medicine professionals should ensure that their players undergo comprehensive rehabilitation and return-to-play programs with utmost fidelity following time-loss injuries.

To our knowledge, this is the first study to report the prevalence of preseason MSK pain in student athletes; hence, the comparison of our findings with previous literature was a challenge. Similar studies in other populations used different methodologies. A study on middle-school student athletes that included soccer and basketball players assessed the frequency of the prevalence of MSK pain over time and reported a range from 19% of infrequent pain to 51% of frequent pain among the study participants [14]. Another study that assessed MSK pain in elite marathon runners reported a 12-month prevalence of 75% [15]. Our finding of a preseason point prevalence of 26% is consistent with the trend that MSK pain is common among competitive athletes. Our finding that one in four players had preseason MSK is clinically important and calls for action. This finding suggests the need for a routine preseason evaluation of MSK pain in collegiate soccer and basketball players, particularly back and knee pain.

Although self-report measures have inherent limitations, including the limitation of recall bias, self-reporting was deemed to be the best MSK pain outcome measure in this study. This was because our outcome of interest was pain and not injury. Participants were also interrogated about current pain and not pain felt in the past. This method was, therefore, appropriate and it was in alignment with the best approaches for studies relating to non-time-loss complaints/problems in competitive athletes [3,4,16].

This study had a few limitations. First, this study had a cross-sectional design. Although this design fitted well with the measures of prevalence, which was the primary objective of this study, it was very limited in assisting our understanding the etiology of preseason MSK pain among players. Our findings regarding the factors associated with preseason MSK pain should be considered with caution when interpreted for clinical use. Nevertheless, this study has provided preliminary information for future research considering the further understanding of MSK pain in collegiate athletes. Future studies should conduct more rigorous studies such as prospective cohort and randomized controlled studies to understand the natural history of MSK pain among student athletes.

A second limitation was that we examined a limited number of potential risk factors for MSK pain. One example of a potential risk factor missing in our multivariable logistic regression model is workload. There is emerging evidence showing that sport-related loads, whether internal or external loads, are related to both MSK injuries and pain, especially overuse pain [11,17,18]. One of the RICHLoad Project objectives is to evaluate this relationship using a prospective design – this is ongoing.

A third limitation related to the instruments used in collecting the study variables. The primary independent variable in this study—injury history in the previous year—was recorded through self-reporting. The accuracy of this method of capturing retrospective data is potentially threatened by recall bias among athletes. However, we believe that there was a much lesser to possibly no impact of recall bias on the additional question regarding their feeling of a completely healed injury or not.

## 5. Conclusions

In this study, one in four collegiate soccer and basketball players had preseason MSK pain and the most frequent body locations were the back and knee. The results of this current study suggest an association between previous injuries and preseason MSK pain. Collegiate sports medicine professionals should consider conducting routine preseason evaluations of MSK pain in collegiate soccer and basketball players and initiate appropriate interventions for the primary and secondary prevention of MSK pain among athletes.

## Figures and Tables

**Table 1 ijerph-19-09128-t001:** Participant characteristics.

	Male (*n* = 42)	Female (*n* = 55)
Age	20.8 (1.75)	19.5 (1.2)
Height, cm	184.9 (9.1)	166.3 (12.0)
Weight, kg	84.1 (12.9)	69.4 (14.3)
Body mass index, kg/m^2^	24.5 (2.3)	24.6 (3.9)
Previous injury, *n* (%)		
No	24 (57.1)	29 (52.7)
Yes, with complete healing	13 (31.0)	16 (29.1)
Yes, with incomplete healing	5 (11.9)	10 (18.2)
Sport (soccer vs. basketball players) *n* (%)	15 (35.7)	36 (65.5)

Values are the mean (standard deviation) unless otherwise indicated.

**Table 2 ijerph-19-09128-t002:** Location of musculoskeletal pain in collegiate soccer and basketball players.

	Soccer (*N* = 51)	Basketball (*N* = 46)	Overall (*N* = 97)
	*n* (%)	*n* (%)	*n* (%)
Back	3 (5.9)	3 (6.5)	6 (6.2)
Knee	3 (5.9)	2 (4.3)	5 (5.2)
Groin	4 (7.8)	-	4 (4.1)
Quadriceps	4 (7.8)	-	4 (4.1)
Ankle	2 (3.9)	1 (2.2)	3 (3.1)
Hamstring	2 (3.9)	-	2 (2.1)
Foot	-	2 (4.3)	2 (2.1)
Upper limb	-	2 (4.3)	2 (2.1)
Chest	-	2 (4.3)	2 (2.1)
Head/neck	1 (2.0)	-	1 (1.0)

*N*: players; *n*: pain location. The total count for pain location was > 25 (individual count of having at least one report of pain prevalence) because several athletes had more than one pain location.

**Table 3 ijerph-19-09128-t003:** Multivariable logistic regression model examining the variables associated with preseason musculoskeletal pain in collegiate soccer and basketball players.

Variable	OR (95% CI)	*p*-Value
Previous injury		
No (referent)	1	
Yes, with complete healing	0.53 (0.19–1.52)	0.240
Yes, with incomplete healing	3.50 (1.28–9.36)	0.014 *
Sex		
Female (referent)	1	
Male	1.12 (0.64–1.98)	0.694
Sport		
Basketball (referent)	1	
Soccer	1.82 (0.73–4.53)	0.196

Final model was controlled for team clusters. Initial model included age and BMI. CI: confidence interval; OR: odds ratio. * Statistical significance at alpha = 0.05.

## Data Availability

Data are available upon reasonable request.

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
