# Peer review of "Absence of Injury Is Not Absence of Pain: Prevalence of Preseason Musculoskeletal Pain and Associated Factors in Collegiate Soccer and Basketball Student Athletes"

_ijerph, 2022, doi:10.3390/ijerph19159128_

Round 1

Reviewer 1 Report

1.0 In the introduction there is discussion as to to the risk of injury recurrence post prior trauma, and to etiology of recurrent injury. I did not see consideration given to altered motor control post injury as a prognostic factor for subsequent reinjury. Though the research on this predominantly related to altered motor control post concussion as a risk factor for ongoing injury, it may be worth consideration. https://www-ncbi-nlm-nih-gov.dml.regis.edu/pmc/articles/PMC7987572/ (section 3 relates to the concept of altered motor control and injury risk such as ACL injury).

2.1 - It would help if the defining characteristics of incomplete healing were outlined for the questionnaire. Was this the presence of ongoing pain, a lack of return to full function, other characteristics? 

3.2 - the calculation for the presence of pre-season pain appears to be based upon the whole cohort (N) with then the percentages for body regions being calculated only from those who had pre-season pain. This presents two different data sets with some confusion as the back pain at 19.4% across all participants is a lower percentage than the sub-groups (21/25% etc.).

3.3 - I am presuming from this description of the analysis that pain was not a component in the definition of incomplete healing, otherwise using pain as a part of the identification of incomplete recovery/healing as a predictor for MSK pain pre-season would be problematic. This needs to be clarified.

Table 2 - did the back include thoracic/lumbar and pelvic?

Discussion - this study is presenting data relating to individuals intending unrestricted participation in soccer/basketball correct? Thus the prevalence of pain in those not in active rehabilitation? 

It is a strong statement to call for action with a total N of 6 individuals with back pain. Yes, this needs to be considered and studied further, and ongoing study taken to identify if there is a correlation of pre-season MSK pain and injury recurrence (if not managed or otherwise). 

It is not clear how the AT or PTs could not determine if an injury was fully healed but self-report could. I infer this to mean that the records would be reviewed but the clinicians not interviewed. I would encourage being less conclusive about this.

Conclusion - is the rate of back and knee pain statistically different from quadriceps/groin? If these are combined to be thigh/pelvis then there is a higher rate of pre-season pain in the thigh/pelvic region. As the N is relatively low and the numbers are 6/5/4/4, focus only to back and knee from the data needs to be lessened in the discussion and conclusion from my view.

Author Response

REVIEWER 1

1.0 In the introduction there is discussion as to to the risk of injury recurrence post prior trauma, and to etiology of recurrent injury. I did not see consideration given to altered motor control post injury as a prognostic factor for subsequent reinjury. Though the research on this predominantly related to altered motor control post concussion as a risk factor for ongoing injury, it may be worth consideration. https://www-ncbi-nlm-nih-gov.dml.regis.edu/pmc/articles/PMC7987572/ (section 3 relates to the concept of altered motor control and injury risk such as ACL injury).

Response: Thank you for this suggestion. We have expanded the introduction to include this additional discussion based on the reference advised.

“One potential mechanism is altered motor control following an initial traumatic sport injury. Research in this area is however currently limited to recurrent injury risk after an incidence of concussion [12].”

2.1 - It would help if the defining characteristics of incomplete healing were outlined for the questionnaire. Was this the presence of ongoing pain, a lack of return to full function, other characteristics? 

Response: We have added the specific questions to clarify your concern.

“The following two questions, from the baseline questionnaire, were used to obtain the response categories for previous injury, study’s primary independent variable: “Have you had any injury requiring at least 1 day of missed participation from sport or exercise in the past 12 months?” and (if yes) “Do you feel your injury/injuries has/have healed 100%?””

3.2 - the calculation for the presence of pre-season pain appears to be based upon the whole cohort (N) with then the percentages for body regions being calculated only from those who had pre-season pain. This presents two different data sets with some confusion as the back pain at 19.4% across all participants is a lower percentage than the sub-groups (21/25% etc.).

Response: Thank you so much for this very crucial observation. The calculations for proportions on Table 2 have now been revised based the entire cohort.

3.3 - I am presuming from this description of the analysis that pain was not a component in the definition of incomplete healing, otherwise using pain as a part of the identification of incomplete recovery/healing as a predictor for MSK pain pre-season would be problematic. This needs to be clarified.

Response: Pain was not a factor. We have clarified this in the manuscript, and we have added the specific questions.

Table 2 - did the back include thoracic/lumbar and pelvic?

Response: We were not able to determine that. Location of pain in the body was based solely on participants’ report.

Discussion - this study is presenting data relating to individuals intending unrestricted participation in soccer/basketball correct? Thus the prevalence of pain in those not in active rehabilitation? 

Response: Yes, that is correct. We have borrowed some of your words to emphasize that in our opening sentence.

“Our primary objective was to assess the prevalence of preseason MSK pain in collegiate soccer and basketball players intending unrestricted participation in their sport.”

It is a strong statement to call for action with a total N of 6 individuals with back pain. Yes, this needs to be considered and studied further, and ongoing study taken to identify if there is a correlation of pre-season MSK pain and injury recurrence (if not managed or otherwise). 

Response: We have added the other body locations to the list.

“The back region was the most common location of pain among players. Other common body locations included the knee, groin and quadriceps.”

It is not clear how the AT or PTs could not determine if an injury was fully healed but self-report could. I infer this to mean that the records would be reviewed but the clinicians not interviewed. I would encourage being less conclusive about this.

Response: We agree with you, we have removed the ambiguous sentence from the paragraph.

Conclusion - is the rate of back and knee pain statistically different from quadriceps/groin? If these are combined to be thigh/pelvis then there is a higher rate of pre-season pain in the thigh/pelvic region. As the N is relatively low and the numbers are 6/5/4/4, focus only to back and knee from the data needs to be lessened in the discussion and conclusion from my view.

Response: Your view is relevant. We have amended our conclusions to be less assertive given the study limitations.

“In this study, one in four collegiate soccer and basketball players had preseason MSK pain and the most frequent body locations were the back, knee, groin and quadriceps.”

Reviewer 2 Report

Dear authors, thank you for your effort in your research. Although your research is a subject that has not been studied much, it contains major errors on some issues. I will reevaluate your work after the corrections I have mentioned below.

abstract

Please write the characteristics of the subjects in the method section, gender and branch (for example 43% male, 57% female).

Conclusion: in this section, just try to advise about pre-season MSK pain for football players and basketball players. At the same time, focus on your results rather than recommendations.

Introduction

In this section, you mentioned the limitations of the studies on MSK pain and exemplified it with a few studies. However, although many studies do not directly include MSK pain, they present findings that we can refer to this issue in their measurements (pain questionnaires, etc.). Please strengthen the originality of your current research by adding a few of these studies.

I recommend that you present the information of the subjects you presented in the Results section in the method section. Because the method part seems to be missing a lot.

Discussion

Please try to support your current findings by taking examples from research (pain questionnaires etc) that examines returning to sports after injury, even if it is not directly MSK pain. As you know, there are many studies that examine especially pre-season injury situations and the factors that reveal these situations.

After all the necessary corrections, I think your research can be published in the journal ijerph. With my most sincere regards.

Author Response

REVIEWER 2

Dear authors, thank you for your effort in your research. Although your research is a subject that has not been studied much, it contains major errors on some issues. I will reevaluate your work after the corrections I have mentioned below.

Thank you for your time in reviewing our manuscript. We have addressed your concerns below and in the revised manuscript as well. The other reviewer also raised several important issues and we have addressed all.

abstract

Please write the characteristics of the subjects in the method section, gender and branch (for example 43% male, 57% female).

Response: We are at capacity for word limit for the abstract, so we are not able to add more words. We believe that readers can intuitively decipher the remaining proportion out of 100(%) based on the stated values.

Conclusion: in this section, just try to advise about pre-season MSK pain for football players and basketball players. At the same time, focus on your results rather than recommendations.

Response: We have amended our conclusion to be less assertive; however, we believe the recommendations are consistent with our results, given the prevalence values found.

“One in four collegiate soccer and basketball players ‘had’ preseason MSK pain.”  

Introduction

In this section, you mentioned the limitations of the studies on MSK pain and exemplified it with a few studies. However, although many studies do not directly include MSK pain, they present findings that we can refer to this issue in their measurements (pain questionnaires, etc.). Please strengthen the originality of your current research by adding a few of these studies.

Response: Relevant literature were highlighted in our discussion. We have reviewed existing literature further and added some details.

I recommend that you present the information of the subjects you presented in the Results section in the method section. Because the method part seems to be missing a lot.

Response: We have added details to the methods section. Please see track changes.

Discussion

Please try to support your current findings by taking examples from research (pain questionnaires etc) that examines returning to sports after injury, even if it is not directly MSK pain. As you know, there are many studies that examine especially pre-season injury situations and the factors that reveal these situations.

Response: Thank you for your suggestion, we have added more literature to the introduction.  

After all the necessary corrections, I think your research can be published in the journal ijerph. With my most sincere regards.

Response: Thank you for your time.

Round 2

Reviewer 2 Report

Dear Authors,

Thank you for your effort. Congratulations!